# “My Brain Can Stop”: An ERP Study of Longitudinal Prediction of Inhibitory Control in Adolescence

**DOI:** 10.3390/brainsci11010100

**Published:** 2021-01-13

**Authors:** Tzlil Einziger, Mattan S. Ben-Shachar, Tali Devor, Michael Shmueli, Judith G. Auerbach, Andrea Berger

**Affiliations:** 1Department of Psychology, Ben-Gurion University of the Negev, Be’er Sheva 8410501, Israel; matanshm@post.bgu.ac.il (M.S.B.-S.); devort@post.bgu.ac.il (T.D.); mickey.shmueli@gmail.com (M.S.); judy.auerbach@gmail.com (J.G.A.); andrea@bgu.ac.il (A.B.); 2Zlotowski Center for Neuroscience, Ben-Gurion University of the Negev, Be’er Sheva 8410501, Israel

**Keywords:** ADHD, effortful control, familial risk, inhibitory control, ERP, N2

## Abstract

We examined the longitudinal predictors of electrophysiological and behavioral markers of inhibitory control in adolescence. Participants were 63 adolescent boys who have been followed since birth as part of a prospective longitudinal study on the developmental pathways to attention-deficit hyperactivity disorder (ADHD). At 17 years of age, they completed the stop-signal task (SST) while electroencephalography (EEG) was continuously recorded. Inhibitory control was evaluated by the stop-signal reaction time (SSRT) as well as by the amplitude of the event-related potential (ERP) component of N2 during successful inhibition. We found that higher inattention symptoms throughout childhood predicted reduced amplitude (i.e., less negative) of the N2 in adolescence. Furthermore, the N2 amplitude was longitudinally predicted by the early precursors of child familial risk for ADHD and early childhood temperament. Specifically, father’s inattention symptoms (measured in the child’s early infancy) and child’s effortful control at 36 months of age directly predicted the N2 amplitude in adolescence, even beyond the consistency of inattention symptoms throughout development. The SSRT was predicted by ADHD symptoms throughout childhood but not by the early precursors. Our findings emphasize the relevance of early familial and temperamental risk for ADHD to the prediction of a later dysfunction in inhibitory control.

## 1. Introduction

Attention-deficit hyperactivity disorder (ADHD) is early-onset and a chronic neurodevelopmental disorder [1], with an estimated prevalence of 5% to 9% among school-age children [2,3]. It is manifested by the behavioral symptoms of inattention, hyperactivity, and impulsivity, which interfere with day-to-day functioning [4]. ADHD is considered to have a substantial genetic component [5,6,7], with heritability ranging from 70% to 80% [5,7]. At the neurocognitive domain, ADHD is associated with deficits in several executive functions [8,9], including the core neurocognitive processes of inhibitory control, working memory, switching, and planning [10], which are essential for goal-directed behavior.

The heterogeneity of ADHD is reflected in both its behavioral symptoms [4] and its neurocognitive deficits [8]. Not all deficits are exhibited by all individuals with ADHD [8,11]; different neurocognitive deficits are assumed to be involved in distinct developmental pathways of ADHD [12]. Therefore, studying the developmental pathways of specific neurocognitive deficits that are associated with ADHD may be useful for the early identification of relatively homogeneous groups of children who may be at risk for particular outcomes.

Among the executive functions, one of the most consistent deficits associated with ADHD is in inhibitory control [9,13]. Inhibitory control refers to the ability to suppress a dominant response that is no longer required or adaptive in favor of subdominant and required response [14,15]. An inhibitory control deficit was estimated to be present in 27% to 50% of children diagnosed with ADHD [8,16], and it appears to be more specifically associated with symptoms of inattention rather than with symptoms of hyperactivity-impulsivity [9,17,18,19,20,21,22]. Importantly, inhibitory control may mark an underlying genetic risk for ADHD, and it is considered a state-independent trait. It persists throughout development, regardless of the overt manifestation of ADHD phenotype, even when it is remitted [23].

### 1.1. Behavioral and Electrophysiological Aspects of Inhibitory Control

At the behavioral level, one of the most widely used measures of inhibitory control is the stop-signal task (SST), which is based on the theoretical horse-race model [24]. The SST addresses stopping a response that has already been prepared for execution. It requires participants to respond to a “go” stimulus (i.e., usually a simple discrimination task) but to withhold their response when the go stimulus is followed by a “stop” signal. The efficiency of the ability to stop the response is considered an index for inhibitory control. However, the latency of this process cannot be directly measured because successful inhibition results in the lack of a behavioral response. The uniqueness of the SST is that it enables estimation of this covert latency, as reflected in the stop-signal reaction time (SSRT) [24,25].

The association between behavioral performance in the SST and ADHD has been well established [23,26,27] in both clinical [27] and nonclinical samples [21,26]. According to a meta-analysis, individuals with an ADHD diagnosis showed a medium-sized inhibitory control deficit, reflected in longer SSRT, compared to control participants [27]. Studies that examined the relation of the SSRT with ADHD subtypes or symptom domains suggested specificity for inattention symptoms, at least among school-age children [19,20,21]. For example, children with symptoms of both inattention and hyperactivity-impulsivity, or inattention alone, had longer SSRT than control participants; however, the performance of children with hyperactivity-impulsivity alone was similar to the performance of the control group [20]. In contrast, in a sample of preschool children, the SSRT was found to correlate with concurrent parental reports of hyperactivity-impulsivity symptoms, but not inattention symptoms, possibly because, at this young age, parents are more sensitive to symptoms of hyperactivity-impulsivity rather than to symptoms of inattention, which become more salient with the entrance to formal schooling [28].

At the neurophysiological level, inhibitory control can be studied with a high degree of temporal resolution by recording event-related potentials (ERPs). This technique enables the identification of the precise timing of brain processes during cognitive tasks; this is especially beneficial in the case of inhibitory control, considering the absence of a behavioral response. The inhibitory control process in the SST has been mainly linked to the ERP component N2 [29], which is a negative deflection in the waveform peaking around 200–250 ms over the right frontal areas, after the onset of the stop signal [30]. An inhibitory control deficit in ADHD has been repeatedly found to be accompanied by a reduced amplitude of the N2 component among individuals with ADHD on successful inhibition trials [28,30,31,32,33,34,35,36]. Another ERP component that is sometimes examined in stop-signal studies related to ADHD is the P3, also referred to as the NoGo-P3 [30,32,33,34,36]. The P3 is a positive deflection in the waveform peaking around 300–500 ms after the stop signal and displaying maximal amplitude over frontocentral regions [30]; it was also found to be reduced among individuals with ADHD [31,33,36]. Although both the N2 and the P3 have been suggested as indices of inhibitory control [37,38], the N2 has also been interpreted as an index of conflict monitoring [39,40]. The P3 has been interpreted as an index of deficits in later stages of stimulus evaluation, occurring after the inhibition process, or as an index of deficits in more general processes of cognitive control [34,36,37,41,42]. It should also be mentioned that earlier ERP components have been found to differ between ADHD and controls [43], but these differences are not directly related to inhibitory control in the SST. In our longitudinal study, when the children were tested with the SST at age 5 years, we found that only the N2 amplitude was related to child ADHD symptoms [28]. Therefore, in the present study, we focused on the N2 as the neural signature of the inhibitory control process. Imaging studies in adults indicate that the prefrontal cortex, and especially the right inferior frontal cortex, are activated in tasks that demand inhibitory control (such as the SST and the Go/NoGo task) and are critically involved in the process of inhibitory control [44].

### 1.2. Early Precursors of an Inhibitory Control Deficit in ADHD

Although ADHD is considered a developmental disorder, its developmental course is not fully known, and, specifically, the developmental pathways leading to its core deficit of inhibitory control in adolescence are relatively uninvestigated. Adolescence is a period of rapid changes in brain maturation, accompanied by extensive behavioral and cognitive changes [45,46], which may be expressed in suboptimal decision-making, high impulsivity, and risk-taking behavior [45]. The intensity of these behaviors may be substantially enhanced in adolescents with ADHD symptomatology [47,48], especially among those with impairments in the executive functioning domain and inhibitory control [47,49]. Indeed, poor inhibitory control in this developmental period was found to be associated with a wide range of maladaptive outcomes, such as substance abuse [50], delinquency [51], and more. Understanding the precursors of a developmental pathway leading to a dysfunction in the behavioral and neural mechanism of inhibitory control is, therefore, important in the context of ADHD, but also for a broader context of other plausible negative developmental outcomes. Identifying the early precursors of this deficit requires a longitudinal design, and more specifically, a “high-risk” strategy. Such a strategy enables the identification of individuals at risk for the disorder—before they manifest any specific deficits or symptoms—and the opportunity to track them throughout development. This would eventually allow a test of the utility of these early precursors as predictors of a certain deficit or the manifestation of the disorder itself [28,52]. However, longitudinal designs are very costly, both in terms of time and effort, and, therefore, such studies are few [28,53,54].

Evidence exists to support a relation between a familial risk for ADHD and child inhibitory control deficits [28,55]. An inhibitory control deficit, reflected in longer SSRT, was found to share genetic risk with ADHD symptoms [26]. Poorer inhibitory control was documented among individuals at genetic risk for ADHD [56], as well as among parents of ADHD probands and non-ADHD siblings, compared to controls [57,58]. Familial risk for ADHD, as reflected by maternal ADHD symptoms, was found to be concurrently related to child SSRT, with a medium-sized effect [55]. In our prospective longitudinal study of boys at familial risk for ADHD, we previously found that paternal inattention symptoms, measured a few months after the birth of the child, predicted child SSRT at 5 years of age [28].

Furthermore, in our longitudinal study, 24-month-old toddlers at high familial risk, based on their fathers’ ADHD symptoms, were rated by their parents as lower on temperamental effortful control (including inhibition, appropriate allocation of attention, and interest) compared to a matched control group [59]. Indeed, such innate individual differences in child temperament can be detected very early in childhood [60,61], and at extreme levels, they may act as precursors to developmental pathways leading to later cognitive deficits and ADHD [52,62,63]. Effortful control is the temperamental aspect related to self-regulation; it refers to relatively deliberate processes, such as attentional (i.e., attentional focusing and shifting) and inhibitory control capacities [61]. It develops throughout childhood and adolescence, and it depends on the maturation of executive aspects of attention [64,65]. The relation between early temperamental effortful control and ADHD symptoms, especially inattention, was well documented in both cross-sectional [66,67] and longitudinal studies [52,54].

A few studies have examined the association between child temperamental effortful control and the N2 amplitude on inhibition trials in the Go/NoGo task [53,68,69,70] or in response to a conflict in the flanker task [68,71]. For example, in a study on attentional training, it was found that children with high levels of temperamental effortful control showed greater N2 amplitude (i.e., more negative) on incongruent trials of the flanker task, compared to children with low levels of effortful control [71]. Hoyniak and colleagues [69] found that among the different aspects of temperamental effortful control, only temperamental inhibitory control was concurrently associated with the N2 amplitude on inhibition trials, measured in typically developed toddlers. Other studies demonstrated that among the components of effortful control, attentional capacities (i.e., attentional focusing or shifting) were associated with the N2 component [68,70]. For example, attentional shifting—but not other aspects of temperamental effortful control—was negatively correlated with the N2 amplitude among school-age children, even after controlling for ADHD symptomatology [70].

Familial risk for ADHD, reflected by parental ADHD symptoms [28,55], and aspects of child effortful control [53,68,69,70] have been associated with the behavioral and electrophysiological markers of inhibitory control. These findings were mostly based on cross-sectional studies that focused on the childhood period [68,69]. Therefore, the utility of these plausible early precursors to the outcome of inhibitory control in later periods of development is unknown. Furthermore, apart from one study [70], the relation between effortful control and the N2 component was not examined in the context of ADHD [53,69]. One of the challenges in examining the predictability of these early precursors to later inhibitory control is that the same precursors were also found to be associated with the behavioral symptoms of ADHD [52,55,67]. It is unclear whether the early precursors are directly related to the neural dysfunction associated with inhibitory control or whether they are related to the exhibition of symptoms during development, and it is these symptoms that predict later inhibitory control. Therefore, in order to assess the unique contributions of familial and temperamental risk to the specific outcome of an inhibitory control dysfunction, it is necessary to control ADHD symptoms throughout development.

### 1.3. The Present Study

This study is part of a prospective longitudinal high-risk study on the developmental course of ADHD and its core neurocognitive deficits. We focused on the behavioral and electrophysiological aspects of inhibitory control in adolescence from a developmental perspective. Our first goal was to corroborate previous findings regarding the specificity of inattention symptoms to behavioral aspects of inhibitory control and to test whether these symptoms, measured several times throughout development, would also show specificity to the electrophysiological aspect of inhibitory control in adolescence. Secondly, we aimed to examine the contribution of familial and temperamental risk for ADHD (i.e., parental ADHD and child effortful control) as early predictors of inhibitory control in adolescence. Based on the literature and previous results from our longitudinal study [28], we hypothesized that: (1) behavioral and electrophysiological markers of inhibitory control in adolescence (i.e., SSRT and the N2 amplitude) would be concurrently and longitudinally predicted by inattention symptoms, rather than by hyperactivity-impulsivity symptoms and (2) higher parental inattention symptoms and lower levels of early childhood temperamental effortful control would predict longer SSRT and reduced N2 amplitude in adolescence. Given that both parental symptoms and early childhood effortful control were found to predict child/adolescent ADHD symptoms [52,55,67,72], we aimed to examine whether these early precursors would have a direct effect on the prediction of inhibitory control, above and beyond the development of child symptoms, or whether they would be indirectly associated with later inhibitory control, only through child inattention symptoms during development.

## 2. Method

### 2.1. Participants

Our sample consisted of 63 male adolescents (*M* = 17.37 years, *SD* = 0.41, range 16.52–18.48) who took part in the Ben-Gurion Infant Developmental Study (BIDS) from birth. Recruitment to the prospective longitudinal study was conducted at the maternity ward of the Soroka Medical Center in Be’er Sheva, Israel. Families were recruited based on several inclusion criteria. First, because of the higher prevalence of ADHD among males, compared to females [73], only families of male newborns were recruited. In addition, all infants who entered the study were born healthy, with normal birth weight (*M* = 3296.07 g, *SD* = 419.75) and gestational age (*M* = 39.19 weeks, *SD* = 1.59). The study included two-parent families; parents were either native-born or immigrants who studied in the country and spoke the local language. At the entrance to the longitudinal study, the mean age of parents was 29.95 years (*SD* = 4.90) for mothers and 33.65 years (*SD* = 5.44) for fathers. The mean number of years of education was 12.80 (*SD* = 1.72) for mothers and 12.32 (*SD* = 1.77) for fathers. Another inclusion criterion was based on paternal ADHD symptoms; at the hospital, fathers completed a yes/no format questionnaire that included 18 ADHD items from the Diagnostic and Statistical Manual of Mental Disorders, 4th edition (DMS-IV) [73]. In the initial phases of the study, children were assigned to either a risk group (i.e., seven or more positive responses) or a comparison group (i.e., three or fewer positive responses), based on this symptoms-count assessment. When children were 2–6 months of age, a more comprehensive assessment of child familial risk was conducted; ADHD symptomatology of both parents was assessed by self- and spousal reports of the Conners’ Adults ADHD Rating Scale [74]. These continuous measures of parental symptoms were used in our study as a measure of familial risk. Our sample included participants who took part in five-time points during the study (at age 2–6 months, 36 months, 54 months, 7 years, and 17 years). This sample did not differ from the rest of the original sample (*Ns* = 50–114, including participants who discontinued participation or who could not be reached) in any of the study variables (including birth weight, gestational age, parents’ education, parental ADHD symptoms, child effortful control, and child ADHD symptoms); all *t*s < |1.66|, *p*s > 0.09.

According to mothers’ reports, 20 (32%) participants were diagnosed with ADHD, at an average age of 8.76 years, by qualified medical professionals. A diagnostic process was not conducted as part of the study, and, therefore, we did not have accurate information regarding the specific presentation of symptoms (i.e., combined, predominantly inattentive, or predominantly hyperactive/impulsive); furthermore, we did not have information regarding the updated status of diagnosis in adolescence. According to the Conners’ Rating Scales-Revised [75] reported by mothers at the adolescence assessment, 10 (16%) participants showed very elevated scores of ADHD symptoms (T scores higher than 70), 3 (5%) showed elevated scores (T scores of 65-69), 10 (16%) showed high average scores (T scores of 60–64), and 40 (63%) showed average scores of symptoms (T scores of 40–59). Mothers’ reports of child diagnosis were significantly related to the severity of symptom levels in adolescence according to the Conners’ Rating Scales, Cramer’s *V* = 0.40, *p* < 0.05.

### 2.2. Measures

#### 2.2.1. Parental ADHD Symptoms

The Conners’ Adult ADHD Rating Scale (CAARS) [74] was used to assess parental ADHD symptoms when children were 2–6 months of age. This questionnaire included a four-point Likert scale ranging from 0 (the behavior rarely or never occurs) to 3 (the behavior occurs very often). Both parents completed the CAARS regarding themselves (using the long self-report version–CAARS-S:L) and regarding their spouses (using the short observer version–CAARS-O:S). For each parent and each symptom domain, the scores of the CAARS-S:L (e.g., father’s self-report of his inattention symptoms) and CAAR-S:O (e.g., mother’s report on father’s inattention symptoms) were averaged to create a single score. Both the CAARS-S:L and the CAARS-O:S showed high reliability; Cronbach’s alpha for both versions was above 0.79. A detailed description can be found in Berger et al. [28].

#### 2.2.2. Effortful Control in Early Childhood

Child effortful control was measured at 36 months of age using the Children’s Behavior Questionnaire (CBQ) [61]. We used the 94-item short-form version of the CBQ; for each item, mothers rated a behavior of their child (e.g., “Can wait to begin new activities if he or she is asked to”) using a Likert scale ranging from 1 (extremely untrue of your child) to 7 (extremely true of your child). Four of the CBQ subscales formed the effortful-control variable (Cronbach’s α = 0.83): attentional focusing (Cronbach’s α = 0.67), inhibition (Cronbach’s α = 0.63), low-intensity pleasure (Cronbach’s α = 0.68), and perceptual sensitivity (Cronbach’s α = 0.69).

#### 2.2.3. ADHD Symptoms throughout Development

The ADHD Rating Scale-IV [76] was used to assess child ADHD symptoms at 54 months of age. This questionnaire assessed the frequency of ADHD symptoms exhibited by the child in the last 6 months. It contained 18 items; in each one, the mother was asked to rate her child on a Likert scale of 0 (never or rarely) to 3 (very often). Cronbach’s alpha was 0.77 for the inattention subscale and 0.79 for the hyperactivity-impulsivity subscale.

The Conners’ Rating Scales-Revised (CRS-R ([75] was used to assess child/adolescent ADHD symptoms at 7 years and 17 years of age. The Conners’ Parent Rating Scale-Revised (CPRS-R) [75] was completed by mothers for their child at 7 years and 17 years of age. Mothers were asked to rate specific symptoms (e.g., “difficulty doing or completing homework”) exhibited by their child/adolescent in the past month, on a scale of 0 (the behavior rarely or never occurs) to 3 (the behavior occurs very often). Cronbach’s alpha for the inattention and hyperactivity-impulsivity subscales was 0.90 and 0.87 at 7 years of age and 0.90 and 0.81 at 17 years of age, respectively.

ADHD symptom domains at 54 months and 7 years were significantly correlated; *r* = 0.37, *p* < 0.01 and *r* = 0.48, *p* < 0.01 for inattention and hyperactivity-impulsivity, respectively. To create a more reliable variable that reflected the consistency of ADHD symptom domains throughout childhood, we averaged the standardized scores of these two assessments (i.e., at age 54 months and 7 years) for each symptom domain (i.e., inattention and hyperactivity-impulsivity symptoms).

#### 2.2.4. Inhibitory Control in Late Adolescence

During a lab visit at 17 years of age, participants completed a computerized SST [24] while electroencephalography (EEG) was continuously recorded. The go stimulus was the number “2” or letter “Z” (50% each) inside a white square, which was presented after a fixation cross. Participants were instructed to press a numbered key on an S-R box; specifically, to press “1” when the go stimulus was “Z” and “4” when the go stimulus was “2”. They were instructed to respond as quickly and as accurately as possible and not to wait for the stop signal to occur. The go trial disappeared from the screen after a response was made or after a maximal duration of 1000 ms. In 30% of the trials, a visual stop signal (a red square outline) appeared randomly and in different delays after the go stimulus; in these trials, participants were instructed to withhold their response. A blank screen appeared between trials for randomly generated intertrial intervals of 1100–1900 ms. The task consisted of one practice block of 40 trials and three test blocks of 80 trials each; in each test block, there were 56 (70%) go trials and 24 (30%) stop trials. A staircase dynamic-tracking procedure [24] adjusted task difficulty by changing the stop-signal delay (SSD) based on adolescents’ performance. The algorithm was programmed to lock on the SSD, which produced 50% successful inhibition trials. The initial delay was set to 500 ms; after a successful stop trial, the SSD increased by 50 ms (this made the following stop trial more difficult), and after an unsuccessful stop trial, the SSD decreased by 50 ms (this made the following stop trial easier). For the behavioral analysis of the SST, we followed Verbruggen et al.’s guidelines [25]. The SSRT was calculated with the integration method with the replacement of go omissions; this method for the estimation of the SSRT was found to be more reliable and less biased, compared to the more traditional “mean method,” especially when combined with a tracking procedure. In the integration method, the finishing time of the stop process was estimated by the nth RT (reaction time); n equals the number of RTs in the RT distribution of go trials multiplied by the probability of response to a stop signal. For the calculation of the nth RT, all go trials with a response were included (including choice error and premature responses), and go omissions were replaced with the maximum RT of the subject. The SSRT was calculated as the difference between the nth RT of go trials and the mean SSD. The SSRT was not estimated for 14 participants because the assumptions of the horse-race model were violated (e.g., cases in which the mean RT on unsuccessful stop trials was higher than the mean RT on go trials or in cases that the probability of response to a stop signal was lower than 0.25 or higher than 0.75). Descriptive statistics of the SST variables for participants who were included in the analyses are presented in Table 1.

### 2.3. Control Variables

#### 2.3.1. IQ Assessment

A shortened version of the Raven’s Standard Progressive Matrices [77] was used to estimate general intelligence at age 13.5 years. A series of 36 diagrams divided into three sets (C, D, E) from the original version was used. Each diagram presented a black and white matrix with one missing part and eight response options. If the participant made three errors in a row, the set was stopped, and the next set was administered. Intelligence scores were calculated by summing the correct answers.

#### 2.3.2. EEG Recording and Preprocessing

EEG data during the SST were recorded from 128 scalp sites using EGI HydrocCel Geodesic Sensor Net (HCGSN) and system [78]. The electrode impedance level was kept below 40 kΩ, an acceptable level for this system [79]. During recording, all channels were referenced to the Cz electrode, the recording frequency band was constant at 0.01 to 100 Hz, and the sampling rate was 250 Hz. Unfortunately, data from 2 participants were unusable due to technical problems with the recorded files. EEG-data preprocessing was carried out using the EEGLAB toolbox (version 14) [80] operating in the MATLAB environment (version 2017a). Continuous EEG data were first high-pass filtered offline at 0.5 Hz and low-pass filtered at 40 Hz. Data were re-referenced to the average of the channels across the scalp. Data were segmented from 200 ms before stimulus to 800 ms after stimulus; three trial types were segmented in the EEG: successful stop trials, unsuccessful stop trials, and successful go trials. The segmented data were visually inspected for artifacts; trials containing large artifacts and bad channels were manually removed. Next, we conducted an independent component analysis using EEGLAB’s runica function. Components containing artifacts that could have been clearly identified (e.g., blinks, muscle twitches) were subtracted from the data. Then, we used an automated bad channel and artifact detection and replacement method (EEGLAB’s TBT plugin) [81], which was followed by manual verification to ensure the good quality of the data. Bad channels were interpolated based on activity from neighboring channels. Four participants were excluded from the ERP analysis based on the following: extremely low performance on the task (i.e., the probability of response to the go signal was approximately 0.5, or the success rate in the stop task tended to zero; three participants) and low-quality EEG data (i.e., a high number of bad channels; one participant). A total of 57 participants had high-quality data, and their data were used in the ERP analysis; the mean number of trials after preprocessing was 35.16 (*SD* = 4.83, range = 23–49) for successful stop and 144.05 (*SD* = 13.91, range 103–161) for successful go.

#### 2.3.3. Measuring of Stop-ERP

In stop-signal locked ERP, there is some overlap with ERP activity arising from the previous go stimulus; although this activity may be suppressed, it is not entirely eliminated by the averaging process. One approach that can solve this problem is to calculate difference waves in which the go stimulus ERP is subtracted from the stop-signal ERP [82,83]. According to the horse-race model, slower RTs for the go signal are related to successful inhibition, and faster RTs for the go signal are related to failed inhibition [84]. Therefore, go trials were divided based on the participants’ median RT to “slow” and “fast” go trials [85]. We followed the procedures reported in O’Halloran et al. [85] and Palmwood et al. [82] and subtracted the slow-go ERP from the successful stop ERP. This difference wave (i.e., stop minus slow go) was used in further analyses. To evaluate the effect of successful inhibition, the mean amplitudes of the N2 were measured. Based on previous findings on the N2 in the SST [34] and after inspection of the grand average ERP of the difference wave between a successful stop and successful slow go (see Figure 1), a time window was selected around the peak of the N2 (180–250 ms), and the right anterior frontal region of interest was used by averaging six electrodes between the standard sites C4 and F8 [31,34]. The split-half reliability of the N2 was 0.77.

### 2.4. Data Analysis Plan

The first aim of the study was to examine the concurrent and longitudinal relation between inhibitory control and ADHD symptoms. This was examined with a series of regression models that were constructed hierarchically to examine the specificity of child ADHD symptoms domains to the electrophysiological and behavioral measures of inhibitory control in adolescence. The control variable of mother’s education was entered in step 1. As recommended by Brocki and colleagues [18] and Wåhlstedt [22], we entered the two symptoms domains (i.e., inattention and hyperactivity-impulsivity) together at step 2; this enabled us to subtract the high overlap between the two symptom domains and examine their specific effects to the prediction. To examine the second aim and test the predictability of the early precursors of ADHD, above and beyond the contribution of child symptoms, path analyses were used. Direct and indirect effects of the early precursors, through child ADHD symptoms, to the electrophysiological and behavioral measures of inhibitory control were calculated; we used the “lavaan” package of R [86] for this purpose. Based on the directional nature of the predictions, one-tailed significant tests were reported.

### 2.5. Compliance with Ethical Standards

#### 2.5.1. Ethical Approval

All procedures performed in studies involving human participants were in accordance with the ethical standards of the Ben-Gurion University of the Negev research committee and with the 1964 Helsinki declaration and its later amendments or comparable ethical standards. The protocol of the study was approved by the Human Subjects Research Committee (1407-1).

#### 2.5.2. Informed Consent

Informed consent was obtained from all individual participants included in the study.

## 3. Results

### 3.1. Missing Data

Among the 57 participants with ERP data, two (3%) did not have data for father’s symptoms, and six (10%) did not have the CBQ data. The missing pattern was *missing completely at random* (MCAR), as indicated by a non-significant Little’s MCAR test, *χ*^2^(35) = 44.94, *p* = 0.12. Multiple imputations were conducted to handle the missing data for early childhood measures; 20 imputations were used to avoid a decrease in statistical power and to achieve an accurate estimation [87].

### 3.2. Preliminary Analyses

Parental background variables (i.e., parents’ age and education at the time of their child’s birth) and adolescent background variables (i.e., age at the adolescence assessment and intelligence scores measured at 13.5 years of age) did not correlate with any of the dependent variables, *r*s < 0.20, *p*s > 0.15, except for mother’s years of education, which had a marginally significant correlation with adolescents’ N2 amplitude at age 17 years, *r* = −0.25, *p* = 0.06; therefore, it was controlled in further analyses. There were no outliers beyond the range of 3.5 *SD* above or below the average of each variable. Descriptive statistics of SST measures are presented in Table 1, and descriptive statistics of ADHD symptoms and the early precursors are presented in Table 2.

### 3.3. Correlation Analyses

The intercorrelations among study variables are presented in Table 3. Bivariate correlation revealed that the N2 amplitude and the SSRT at 17 years of age were not associated with concurrent inattention symptoms and concurrent hyperactivity-impulsivity symptoms. However, both were significantly associated with ADHD symptoms throughout childhood; those who had consistently high levels of ADHD symptoms throughout childhood showed reduced N2 amplitude (i.e., a less negative amplitude) and a longer SSRT at 17 years of age. Fathers’ inattention symptoms, measured when their children were 2–6 months of age, were significantly associated with adolescent N2 amplitude at 17 years of age (this correlation remained significant when we controlled for fathers’ hyperactivity-impulsivity symptoms); this association was not found for fathers’ hyperactivity-impulsivity symptoms, and it was only at trend-level for mothers’ symptoms. Furthermore, child effortful control, measured at 36 months of age, was significantly associated with the N2 amplitude at 17 years of age; lower temperamental effortful control in childhood was related to reduced N2 amplitude. The SSRT was not significantly correlated with parental symptoms or child effortful control.

### 3.4. Testing the Contribution of ADHD Symptoms to the Behavioral and Electrophysiological Markers of Inhibitory Control

#### 3.4.1. Inhibitory Control in Adolescence and Concurrent ADHD Symptoms

Although the linear correlation between the N2 mean amplitude and concurrent symptoms was not significant, when dividing each symptoms domain into quartiles, there was a main effect for group, *F*(3, 52) = 2.96, MSE = 3.65, *p* < 0.05, η^2^p = 0.14, in the inattention domain. Participants in the lower quartile of inattention symptoms showed larger negative amplitudes than those in the other three quartiles together, *F*(1, 52) = 7.94, MSE = 3.65, *p* < 0.01, η^2^p = 0.13; no significant differences were found between the top three quartiles. No significant effects were found for the quartiles of hyperactivity-impulsivity symptoms. The quartiles that were used to produce relatively equal-sized groups were: T scores for the lower quartile (i.e., low symptomatology), which were 40–47 for inattention (*n* = 14) and 43–49 for hyperactivity-impulsivity symptoms (*n* = 14); T scores for the upper quartile (i.e., high symptomatology) were 59-77 for inattention (*n* = 12) and 68-87 for hyperactivity-impulsivity symptoms (*n* = 18). It should be mentioned that these analyses were conducted using ANCOVAs, controlling for mothers’ education. See Figure 2 for an illustration of the difference in the N2 mean amplitude between adolescents with high concurrent inattention symptoms compared to adolescents with low concurrent inattention symptoms. No significant effects were found relating concurrent ADHD symptoms with SSRT, even when dividing into quartiles.

#### 3.4.2. Predicting Inhibitory Control in Adolescence from Childhood ADHD Symptoms

To examine the longitudinal predictions of childhood symptoms domains (i.e., inattention and hyperactivity-impulsivity symptoms, measured at 54 months and 7 years of age) to the behavioral and electrophysiological markers of inhibitory control at 17 years of age, we computed hierarchical regressions models for the prediction of the SSRT and the N2 mean amplitude measures. The results are presented in Table 4. The control variable of mothers’ education was entered in step 1 and made a significant contribution to the prediction of the N2 amplitude and the SSRT. The two symptoms domains were entered in step 2; inattention symptoms throughout childhood uniquely predicted the N2 amplitude, above and beyond hyperactivity-impulsivity symptoms, and mothers’ education. Hyperactivity-impulsivity symptoms throughout childhood did not contribute to the prediction of N2 amplitude. Inattention and hyperactivity-impulsivity symptoms throughout childhood did not uniquely predict the SSRT. In other words, neither of these symptom domains showed specificity for the prediction of the SSRT (when testing the same model with the total ADHD symptoms scale, total ADHD symptoms throughout childhood significantly predicted the SSRT at 17 years of age, β = 0.28, *p* < 0.05). The models were significant, and adjusted R^2^ was 11%, *F*(3, 51) = 3.24, *p* < 0.05, for the prediction of the N2 amplitude and 10%, *F*(3, 45) = 2.81, *p* < 0.05, for the prediction of the SSRT.

### 3.5. Longitudinal Pathways to the Electrophysiological Marker of Inhibitory Control

A path analysis was constructed to examine the direct paths of early precursors of ADHD (i.e., early childhood effortful control and paternal inattention symptoms) and their indirect paths through childhood inattention symptoms to the mean N2 amplitude in adolescence. As seen in Figure 3, child effortful control had a direct negative path on the prediction of N2 amplitude because lower levels of effortful control were related to reduced N2 amplitude in adolescence (i.e., a less negative amplitude). Paternal inattention had a direct positive path in the prediction of N2 amplitude because higher levels of paternal symptoms were related to reduced N2 amplitude. Both childhood effortful control and paternal inattention significantly predicted child inattention symptoms throughout childhood. However, these symptoms, in turn, did not explain a unique additional part of the variance of the N2 amplitude, above and beyond what was already explained by the early precursors. No indirect paths through inattention symptoms were found; *β* = −0.06, *z* = −0.86, *p* = 0.40 and *β* = 0.03, *z* = 0.82, *p* = 0.41 for childhood effortful control and paternal inattention, respectively. The entire path model explained 27% of the variance of inattention symptoms throughout childhood and 29% of the variance of the N2 amplitude; it showed a good fit to the data with a comparative fit index (CFI) of 1, a goodness-of-fit (GFI) of 0.99, a root mean squared error of approximation (RMSEA) lower than 0.01, and a standardized root mean squared residual indices (SRMR) of 0.02. ERP waveforms and topographic maps illustrating the N2 for adolescents with high and low levels of childhood effortful control and paternal inattention symptoms are shown in Figure 4. A similar path analysis was also constructed with concurrent inattention symptoms, and results were similar; child effortful control and paternal inattention symptoms had significant direct paths to the prediction on the N2 amplitude, *β* = −0.33, *z* = −2.71, *p* < 0.01 and *β* = 0.35, *z* = 2.94, *p* < 0.01, respectively. The early precursors did not predict the SSRT in adolescence.

### 3.6. Additional Analyses

To examine which aspects of child temperamental effortful control predicted the N2 amplitude, we conducted similar path analyses, as presented in Figure 3, but instead of the effortful control factor, we entered one of the individual subscales comprising this factor (i.e., attentional focusing, inhibition, low-intensity pleasure, and perceptual sensitivity) into each model. We found that above and beyond the control variable of mother’s education and inattention symptoms, only attentional focusing had a direct path to the N2 amplitude in adolescence, *β* = −0.29, *z* = 2.22, *p* < 0.05. The subscales of inhibition, low-intensity pleasure, and perceptual sensitivity did not predict the N2 amplitude in adolescence.

## 4. Discussion

Using a prospective longitudinal design, we examined the utility of early childhood familial and temperamental risk factors of ADHD as predictors of behavioral and electrophysiological markers of inhibitory control in adolescence. First, we found that a reduced amplitude of the N2 component during successful inhibition (measured from the differences between the successful stop and successful slow go trials) was predicted by higher levels of inattention symptoms throughout childhood and not by hyperactivity-impulsivity symptoms. Childhood ADHD symptoms predicted the SSRT, but the effect was not specific to inattention or hyperactivity-impulsivity symptoms. Second, we found that paternal inattention symptoms, measured a few months after the child’s birth, and child temperamental effortful control, measured at 36 months of age, directly predicted individual differences in the N2 amplitude of adolescents. Direct pathways of predictions were found above and beyond the indirect pathway via the development of inattention symptoms. Adolescents’ SSRT was not predicted by the early precursors. These findings and their interpretations are discussed in detail.

As a first stage, we tested the basic association between the behavioral and electrophysiological markers of inhibitory control and ADHD symptom domains; this was tested separately for concurrent and childhood symptoms. We found that higher levels of inattention symptoms throughout childhood were associated with reduced N2 amplitude during successful inhibition in adolescence. Such a linear correlation was not found with concurrent symptoms; however, the adolescents with the lowest level of inattention symptoms displayed larger N2 amplitudes compared to the rest. Our results were consistent with previous findings on the N2 in the SST, demonstrating a reduced amplitude among children diagnosed with ADHD compared to controls [31,32,33,34]. The results added to such extant literature by demonstrating the relationship between the N2 amplitude and ADHD using a dimensional approach, measuring ADHD symptom domains on a continuum. Our study also provided evidence that inattention symptoms in childhood could longitudinally predict the electrophysiological response associated with inhibitory control in adolescence in a sample of adolescents at varying levels of familial risk for ADHD.

First-order correlations between the N2 amplitude and the SSRT were found with both ADHD symptom domains in childhood. However, after accounting for the high overlap between the two domains, only inattention symptoms were predictive of the N2 amplitude, and neither of them was predictive of the SSRT (indicating the lack of specificity to either of the symptom domains). This emphasized the need to subtract the effect of this overlap when examining the specificity of ADHD symptom domains (as also recommended by [18,22]). The rationale of this specificity to inattention can be understood when considering the relation between the different attentional networks and ADHD; individuals with ADHD show prominent impairments in executive aspects of attention [88,89]. This includes difficulties in target detection, resolution of conflicts, and difficulties in the ability to inhibit distractions or automatic responses [88]. Castellanos and colleagues [90] suggested that neurocognitive functions, such as inhibitory control and working memory (i.e., “cool” executive functions according to the model of Zelazo & Muller [91]), are related to inattention symptoms, whereas delay aversion and other functions involving motivation and affect (i.e., “hot” executive functions ) are related to hyperactivity-impulsivity symptoms [90]. Indeed, such specificity to inattention symptoms has been demonstrated in previous studies using behavioral measures of inhibitory control [18,20,21,22]. Although we did not find this specificity with the SSRT, our study suggested that such specificity exists for the electrophysiological marker of inhibitory control, as measured by the N2 amplitude in the SST.

Interestingly, this specificity was also found for paternal symptoms; paternal inattention symptoms predicted adolescent N2 amplitude, measured almost 17 years later. Our results suggested a plausible parent-of-origin effect in relation to adolescent inhibitory control because it was found that only paternal inattention symptoms, but not maternal symptoms, contributed to the prediction of adolescent N2 amplitude. Such an effect was also found in an earlier assessment in our study because only paternal inattention symptoms were predictive of child SSRT at 5 years of age [28]. Consistently, a relevant parent-of-origin effect was found in Goos and colleagues’ [57] study, which demonstrated that child inhibitory control was more strongly associated with paternal inhibitory control compared to maternal inhibitory control. Results were also consistent with literature indicating that the heritability of ADHD is stronger from fathers to sons, compared to the heritability from mothers to sons [6] and stronger from fathers to sons compared to daughters [92]. In contrast, our findings were not in line with Thissen and colleagues [55], who found an opposite parent-of-origin effect because only maternal symptoms were concurrently related to child SSRT. This contradiction may be related to major differences in the studies’ design and methodology; for example, Thissen and colleagues [55] used a cross-sectional clinical design (with a wide age range) and the SSRT as a measure of inhibitory control; our study was prospective and longitudinal, and its initial recruitment was based on fathers’ symptoms of ADHD. It also included a sample of boys at different levels of familial risk for ADHD and used an electrophysiological measure of inhibitory control at a specific time point. Future research is necessary to fully understand whether fathers’ and mothers’ ADHD could have distinct influences on their children’s neurocognitive functioning, particularly inhibitory control. An understanding of certain parent-of-origin gender and a specific symptom domain could contribute to the early identification of more homogeneous subgroups of children at risk, according to a certain history of parental ADHD.

We also found that temperamental effortful control at 36 months of age was predictive of the N2 amplitude at 17 years of age because lower levels of effortful control were related to a reduced N2 amplitude on successful inhibition. Child effortful control had a direct effect on the prediction of the N2 amplitude; this was above and beyond inattention symptoms (throughout childhood and in adolescence) and child familial risk of ADHD. This means that even after accounting for the part that ADHD (risk or actual symptoms exhibited by the child) explains in the variance of the N2 amplitude, child effortful control still explained a unique part in this variance. This indicated that this relation was not derived from the overlap between effortful control and N2 amplitude with ADHD symptomatology throughout development. Such a longitudinal relation reflected the consistency of deficits in aspects of self-regulation throughout development. This was in accordance with previous cross-sectional studies on typically developing children, which showed concurrent associations between aspects of temperamental effortful control and the N2 amplitude in response to inhibition, using the Go/NoGo task [69], or a conflict, using the flanker task [68,71]. Furthermore, our results suggested that among the different aspects of effortful control, difficulties in attentional focusing, detectable as early as 36 months of age, remained relatively persistent throughout development and could later be expressed in the electrophysiological response associated with inhibitory control. Temperamental attentional focusing might act as an early marker of risk for a developmental pathway leading to inattention symptoms throughout development and to an inhibitory control deficit in adolescence. This was in line with the findings of Wiersema and Roeyers [70] and Buss and colleagues [68] on the relation between the N2 amplitude and attentional aspects of effortful control.

The path analysis demonstrated that both paternal inattention and child effortful control contributed to the prediction of inattention symptoms throughout childhood; however, the contribution of inattention symptoms to the prediction of the N2 amplitude was no longer significant. Inattention symptoms did not explain a unique additional part of the variance, above and beyond the part that was already explained by the early precursors. It should be considered that when we tested the contribution of ADHD symptoms in the more basic model before we added the early precursors, we found a significant modest effect of inattention symptoms throughout childhood. Still, a large part of the N2 variance was not explained by child symptomatology because individuals who exhibit deficits in inhibitory control at the brain activity level do not necessarily exhibit symptoms of ADHD. It seems that the early precursors were able to directly explain the part of the N2 variance that was explained by inattention symptoms, as well as some of the variance that was not explained by these symptoms. It is possible that the genetic and environmental risk, inherent in paternal symptomatology and child temperament, was able to predict inhibitory control difficulties of those who were at some level of risk for ADHD but did not exhibit a high level of inattention symptoms throughout development. This was consistent with evidence suggesting poor inhibitory control among non-affected family members of individuals with ADHD as compared to controls [57,58]. This may suggest that even when the risk for ADHD does not manifest in actual symptoms, it can still be reflected in poorer functioning in the neural mechanism associated with inhibitory control.

In a broader sense, deficits in self-regulatory skills in childhood and adolescence, including inhibitory control, are associated with a wide range of negative outcomes during development [50,51,93,94,95], for example, in poor social, emotional, and cognitive coping [94], poorer physical health and personal finances [93], externalizing problems, substance dependency, and delinquency [50,51,93]. An inhibitory control deficit also appears in other neurodevelopmental disorders, such as autism spectrum disorders and fetal alcohol syndrome, and predicts impairments in executive functioning that may persist throughout development [96,97,98]. Our findings strengthened the results of these previous studies by providing support at the brain activity level, showing that lower levels of early childhood effortful control, especially attentional focusing, could remain persistent throughout development and later be expressed as dysfunctions in the neural mechanism of inhibitory control in adolescence. Alongside previous studies [50,51,93,94,95], our findings emphasized the importance of early detection of children at familial and temperamental risk for ADHD. Such early detection could potentially lead to the early implementation of intervention programs, which could mitigate the plausible negative outcomes associated with poor self-regulatory skills and inhibitory control throughout development.

The main aim of the study was to test the contribution of early familial and temperamental risk for ADHD to the prediction of inhibitory control in adolescence. Because of the overlap between early precursors, ADHD symptoms, and inhibitory control, it was necessary to examine whether the early precursors were directly related to inhibitory control or whether the relation actually derived from the overlap between inhibitory control and ADHD symptomatology. We found that the contributions of the early precursors to the electrophysiological aspect of inhibitory control were direct. Despite the statistical model that was tested and presented in Figure 3, it should be noticed that the design of our study was not suitable to determine the directional causal relation between inhibitory control and ADHD, and we do not argue that ADHD symptomatology is accountable for a later inhibitory control deficit, as the causal direction can be in the opposite direction. Indeed, an inhibitory control deficit has been suggested as an endophenotype or a liability factor of ADHD [23,26,99]; therefore, we can speculate that the genetic and environmental risk factors of ADHD may contribute to the development of an inhibitory control deficit and that such a deficit may, in turn, increases the risk for ADHD.

Although SSRT was found to be related to N2 amplitude and ADHD symptoms in childhood, in contrast with previous behavioral studies [26,27], we did not find significant associations between SSRT, concurrent ADHD symptoms, or the early precursors. It should be considered that to estimate the SSRT, a few assumptions need to be met; in cases of suboptimal performance in a task or a serious violation of the assumptions, the reliability of the SSRT is compromised, and it is, therefore, advisable to refrain from estimating it [25]. Based on this recommendation, we did not estimate the SSRT for 14 participants, which led to a decrease in the sample size and statistical power in all analyses that included the SSRT. It is also possible that the SSRT has less sensitivity (compared to electrophysiological measures) for detecting subtle individual differences in a sample of adolescents at varying levels of risk for ADHD (as compared to the more prominent differences that are usually found when examining diagnosed individuals versus control participants).

Our study has some limitations that should be considered. First, our sample was limited to males. The decision to use a male-only sample was made when the study began, about 20 years ago, based on the higher prevalence of ADHD among males compared to females [73], and to increase the probability of eventually having enough participants who show symptomatology of ADHD. Second, our sample size was modest, which also limited our statistical power. The data were collected over many years, which naturally led to sample attrition. Still, those who remained in the study until adolescence did not differ from those who dropped out in any of the variables included in the present study. Third, the reliance on questionnaires to assess parental ADHD symptoms, child ADHD symptoms, and child temperament is a further limitation; however, it should be noted that we used procedures that could improve their reliability. For example, parental symptoms were assessed using a combination of self and spousal reports, and child ADHD symptoms were assessed at several time points throughout development. Furthermore, our main findings were based on the relations between aspects assessed in questionnaires and adolescents’ electrophysiological activity; therefore, it cannot be argued that a shared-method variance was accountable for the results. It should also be noticed that according to the mother’s reports of adolescents’ ADHD symptoms, there was a low rate of T-scores in the clinical or subclinical range. This limits, to some degree, our ability to generalize our conclusions regarding the risk for ADHD as a disorder. Furthermore, the present paper was purposely focused on the N2. Further study is required to explore earlier and later ERP components and additional electrophysiological signatures, such as the power of theta frequency, intrasubject variability, and inter-trial coherence, etc.

## 5. Conclusions

The uniqueness of the study is its prospective longitudinal design, which enabled us to test how early childhood risk factors of ADHD contributed to the development of the electrophysiological signature of inhibitory control in adolescence. Our results highlight the close association between the brain activity of adolescents during inhibition and the development of attentional capacities on several levels. First, higher levels of inattention symptoms throughout childhood were found to predict a dysfunction in the brain mechanism involved in inhibition, as reflected by a reduced N2 amplitude in adolescence. Beyond that, paternal inattention as well as early childhood effortful control, particularly attentional focusing, predicted this electrophysiological response during inhibition, measured more than a decade later. These factors were also involved in the development of child inattention symptoms, but their predictions of adolescent brain activity were direct. These risk factors may represent an enduring biological risk, which could be related to the brain mechanism involved in inhibition. Our findings emphasize that an early deficit in self-regulatory skills may persist throughout development; early detection of children at risk for inhibitory control deficits could contribute to early intervention programs designed to mitigate future negative outcomes from such a deficit.

## Figures and Tables

**Figure 1 brainsci-11-00100-f001:**
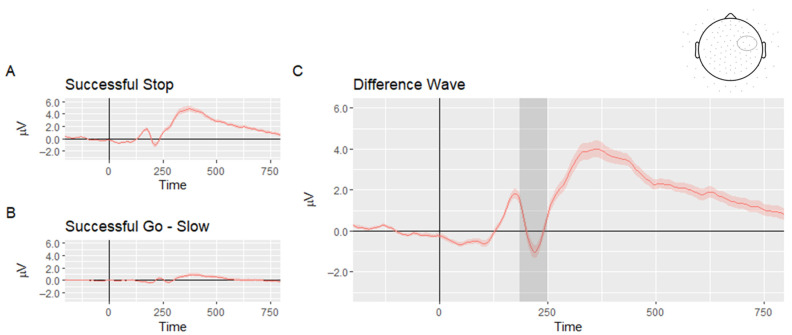
Grand average event-related potential waveforms of (**A**) successful stop (**B**) successful go of slow reaction times and (**C**) difference wave between a successful stop and successful slow go. *Note*. The shaded area represents standard error; the gray area delineates the N2 time window; the region of interest is marked with a circle on the electrode map at the top of the figure.

**Figure 2 brainsci-11-00100-f002:**
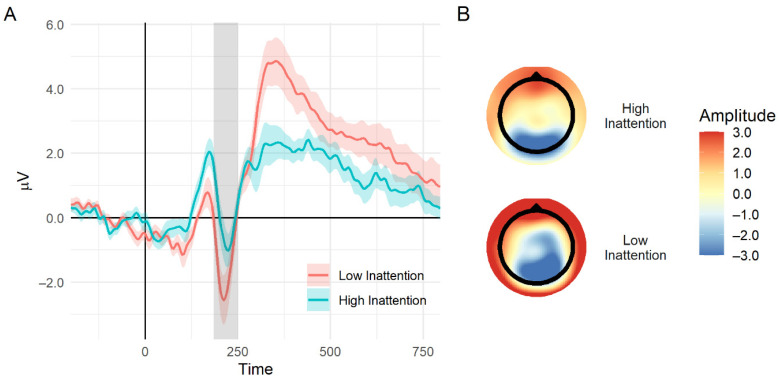
(**A**) An illustration of the event-related potential wave plot of the difference between a successful stop and successful slow go for adolescents with high and low concurrent inattention symptoms (group defined by the lower and upper quartile). The gray area delineates the N2 time window; the shaded area represents standard error. (**B**) Topographic maps of voltages on the scalp at the N2 time window (specifically at 200 ms after stop-signal presentation) for high (top) and low (bottom) concurrent inattention symptoms.

**Figure 3 brainsci-11-00100-f003:**
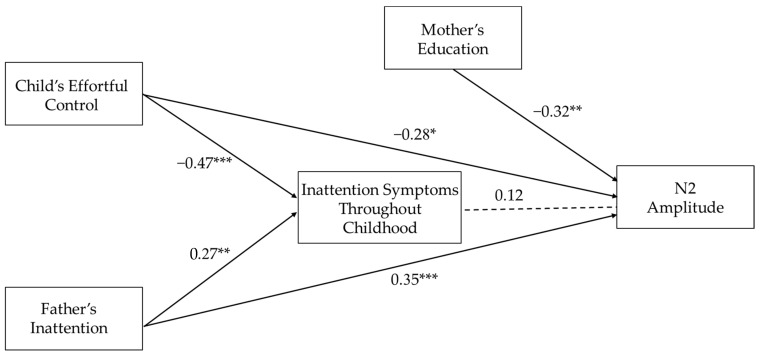
A longitudinal pathway from early risk to N2 amplitude at 17 years of age. *Note:* * *p* < 0.05, ** *p* < 0.01, *** *p* < 0.001; Standardized estimates of the paths were used; significant paths are represented with solid lines, and insignificant paths are represented with dotted lines.

**Figure 4 brainsci-11-00100-f004:**
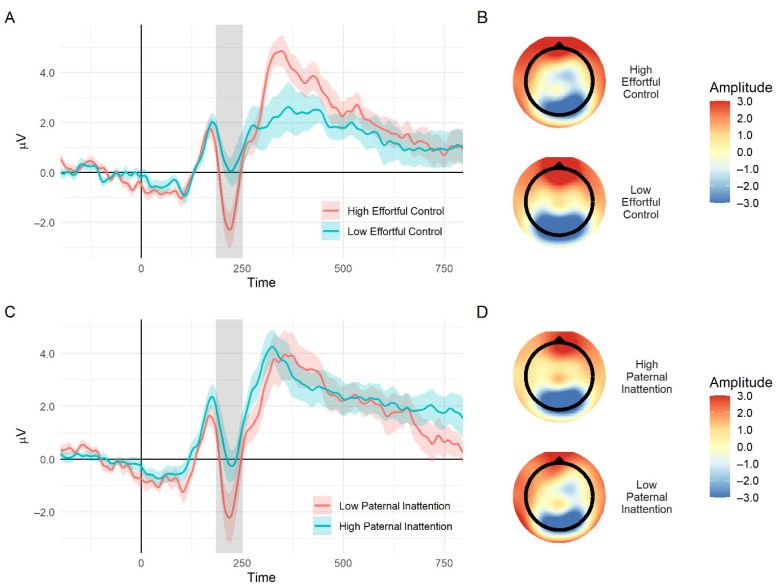
(**A**) An event-related potential wave plot of the difference between successful stop and successful slow go for adolescents with high and low ratings of childhood effortful control (group divided into lower and upper quartiles for illustration purpose). The gray area delineates the N2 time window; the shaded area represents standard errors. (**B**) Topographic maps of voltages on the scalp at the N2 time window (specifically at 200 ms after stop-signal presentation) for high (top) and low (bottom) childhood effortful control. (**C**) An event-related potential wave plot of the difference between a successful stop and successful slow go for adolescents with high and low paternal inattention symptoms (group defined by the lower and upper quartile). (**D**) Topographic maps of voltages on the scalp at the N2 time window for high (top) and low (bottom) paternal inattention.

**Table 1 brainsci-11-00100-t001:** Descriptive Statistics of SST Measures at Age 17 Years.

Variable	Mean (SD)	Range	*n*
Probability of go omissions	0.005 (0.002)	0.003–0.017	49
Probability of choice errors on go trials	0.05 (0.06)	0–0.37	49
Mean RT on go trials	678.49 (165.63)	431.74–1153.65	49
nth RT	576.01 (85.61)	438.80–894.72	49
SD RT on go trials	195.20 (87.05)	67.27–488.11	49
Probability of responding on a stop trial	0.43 (0.07)	0.28–0.55	49
Mean SSD	537.67 (304.33)	202.78–1572.22	49
SSRT	159.90 (56.29)	46.11–278.17	49
N2 mean amplitude	−0.12 (2.06)	−5.87–4.63	57

SST = stop-signal task; RT = reaction time; SD = standard deviation; SSD = stop-signal delay; SSRT = stop-signal reaction time.

**Table 2 brainsci-11-00100-t002:** Descriptive Statistics of ADHD Symptoms throughout Development and Child Risk for ADHD.

	Variable	Mean (SD)	Range	*n*
Concurrent ADHD symptoms	Inattention symptoms T-score—17 y	53.37 (9.28)	40–77	63
	H/I symptoms T-score—17 y	58.56 (12.73)	43–87	63
	Inattention symptoms T-score—7 y	49.10 (7.52)	40–79	59
ADHD symptoms throughout childhood	H/I symptoms T-score—7 y	51.81 (8.22)	41–79	59
	Inattention symptoms—4.5 y	5.16 (3.21)	0–12.00	56
	H/I symptoms—4.5 y	6.73 (3.48)	1.00–16.00	56
Child risk for ADHD	Father’s inattention symptoms	11.31 (5.96)	0–25.00	61
	Father’s H/I symptoms	15.55 (7.59)	2.00–37.00	61
	Mother’s inattention symptoms	10.67 (5.50)	1.00–24.00	61
	Mother’s H/I symptoms	12.72 (6.02)	3.00–27.00	61
	Child’s effortful control—36 months	9.64 (1.55)	6.16–13.66	57

ADHD = attention-deficit hyperactivity disorder; H/I = hyperactivity-impulsivity.

**Table 3 brainsci-11-00100-t003:** Pearson Correlations among Study Variables.

	1.	2.	3.	4.	5.	6.	7.	8.	9.	10.	11.
N2 mean amplitude											
2. SSRT—17 years	0.31 *										
3. Concurrent inattention symptoms	0.14	0.17									
4. Concurrent H/I symptoms	0.17	0.14	0.68 ***								
5. Childhood inattention symptoms	0.28 *	0.23 *	0.50 ***	0.44 ***							
6. Childhood H/I symptoms	0.20 ^+^	0.26 *	0.24 *	0.40 ***	0.53 ***						
7. Father’s inattention symptoms	0.25 *	−0.25	0.12	0	0.14	−0.03					
8. Father’s H/I symptoms	0.17	−0.03	0.01	−0.20	0.12	0.01	0.36 **				
9. Mother’s inattention symptoms	0.20 ^+^	−0.18	0.17+	0.10	0.42 ***	0.10	0.37 **	0.32 *			
10. Mother’s H/I symptoms	0.21 ^+^	−0.19	−0.06	0.02	0.17 ^+^	0.07	0.23*	0.10	0.38 **		
11. Child’s effortful control—36 months	−0.32 *	−0.12	−0.34 **	−0.38 ***	−0.35 **	−0.33 **	0	−0.09	−0.26 *	−0.20+	

^+^*p* < 0.10, * *p* < 0.05, ** *p* < 0.01, *** *p* < 0.001; SSRT = stop-signal reaction time; H/I = hyperactivity-impulsivity.

**Table 4 brainsci-11-00100-t004:** Predicting Inhibitory Control in Adolescence from ADHD Symptoms Domains in Childhood.

	N2 Mean Amplitude		SSRT	
Predictor	β	*R* ^2^	β	*R* ^2^
Step 1				
Mother’s education	−0.25 *	0.06 ^+^	−0.28 *	0.08 *
Step 2				
Child’s inattention throughout childhood	0.31 *	0.16 *	0.14	0.16 *
Child’s H/I throughout childhood	0		0.17	

^+^*p* < 0.10, * *p* < 0.05; ADHD = attention-deficit hyperactivity disorder; H/I = hyperactivity-impulsivity; SSRT = stop-signal reaction time.

## Data Availability

The data presented in this study are available on request from the corresponding author. The data are not publicly available due to privacy and ethical restrictions.

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
