# Peer review of "“My Brain Can Stop”: An ERP Study of Longitudinal Prediction of Inhibitory Control in Adolescence"

_brainsci, 2021, doi:10.3390/brainsci11010100_

Round 1

Reviewer 1 Report

The authors show that one can partly predict adolescent's (n=63 males aged 17y) inhibition-related N2-ERP from a STOP signal task using their earlier and their parental ADHD scores in a longitudinal sample varying in ADHD risk. The right frontal N2 during successful inhibition was found to be attenuated in those with higher father's inattention scores and higher own inattention scores during childhood.

The relatively large prospective longitudinal sample with assessment of parental ADHD and risks and symptoms since birth, and including a low and high ADHD risk group, are clear strengths of this study tracking aspects of the development of temperament, attention and inhibition. Also, the STOP task ERPs is an informative choice to assess inhibitory control, the models testing the role of early precursors and childhood symptoms seems well described and motivated, and the results are important for understanding pathways to variation in behavioral and neural measures of adolescent's inhibitory controls

Major Issues

1. Relation to ADHD - risk group sampling and ADHD status:

Interpretability is limited due to the lack of ADHD scores interpretable in terms of clinical or subclinical cutoffs (such as Conners T-scores presumably reported in Einziger et al 2018) and the proportion of adolescents with actual ADHD diagnoses and impairment. ADHD, risks for ADHD, and the importance of sampling comparison and risks group are prominently implicated (abstract and introduction: "prospective longitudinal study on the developmental pathways to attention-deficit hyperactivity disorder (ADHD)") but the discussion suggests the findings refer to variations in a " nonclinical sample of adolescents".
The link to ADHD as a disorder remains speculative without reporting and discussing the ADHD status or level, and whether this actually differed between the risk and comparison groups. This should be made clear in the abstract and throughout results and discussion. The final sentence that "Our findings have plausible clinical implications because they may contribute to early identification of children at risk of a later IC dysfunction and ADHD." should be toned down unless the authors demonstrate that their sample extends into the clinical range suffereing from dysfunction.

2. Stop task ERPs

Focusing hypotheses on the sucessful stop N2 is fine, but effects on other components, particularly the STOP-P3 but also earlier activity preceding the stop signal, which are also implicated in ADHD in most papers cited (including their own) are needed to support specific interpretions. The ERP illustrations are also not sufficient to allow judging basic ERP data quality and the presence of typical ERP tpographies and effects.

Details

As child ADHD Symptoms at age 54 months, 7 years, and 17 years (i.e current ADHD symptoms) were averaged they contain predictors (implying temporal precedence) but also imple correlates of the N2 at age 17.
Table 1 -check font size
Please spell out inhibitory control (IC is not a common abbreviation in this field)

Author Response

Response to Reviewers

We are very grateful to be given the chance to revise and resubmit this manuscript. We thank the editor and the reviewers for their thoughtful and helpful comments. We have revised the manuscript based on their suggestions.

Reviewer 1:

The authors show that one can partly predict adolescent's (n=63 males aged 17y) inhibition-related N2-ERP from a STOP signal task using their earlier and their parental ADHD scores in a longitudinal sample varying in ADHD risk. The right frontal N2 during successful inhibition was found to be attenuated in those with higher father's inattention scores and higher own inattention scores during childhood.

The relatively large prospective longitudinal sample with assessment of parental ADHD and risks and symptoms since birth, and including a low and high ADHD risk group, are clear strengths of this study tracking aspects of the development of temperament, attention and inhibition. Also, the STOP task ERPs is an informative choice to assess inhibitory control, the models testing the role of early precursors and childhood symptoms seems well described and motivated, and the results are important for understanding pathways to variation in behavioral and neural measures of adolescent's inhibitory controls

Major Issues

  1. Relation to ADHD - risk group sampling and ADHD status:

Interpretability is limited due to the lack of ADHD scores interpretable in terms of clinical or subclinical cutoffs (such as Conners T-scores presumably reported in Einziger et al 2018) and the proportion of adolescents with actual ADHD diagnoses and impairment. ADHD, risks for ADHD, and the importance of sampling comparison and risks group are prominently implicated (abstract and introduction: "prospective longitudinal study on the developmental pathways to attention-deficit hyperactivity disorder (ADHD)") but the discussion suggests the findings refer to variations in a " nonclinical sample of adolescents".
The link to ADHD as a disorder remains speculative without reporting and discussing the ADHD status or level, and whether this actually differed between the risk and comparison groups. This should be made clear in the abstract and throughout results and discussion. The final sentence that "Our findings have plausible clinical implications because they may contribute to early identification of children at risk of a later IC dysfunction and ADHD." should be toned down unless the authors demonstrate that their sample extends into the clinical range suffereing from dysfunction.

  • Thank you for this comment; we have now better clarified the ADHD status of the sample and included the proportions of adolescents at different severity levels according to T scores of the Conners (lines 210-220). The T-scores are included in Table 2 (lines 386).
  • We should mention that we do have some information regarding ADHD formal diagnosis, however, this information is very limited for several reasons; (1) we did not directly assess the participants for ADHD and therefore we could not determine whether the undiagnosed participants were undiagnosed because they did not show symptoms or because they did not undergo a diagnostic assessment of their symptoms; (2) parents did not always know what was the subtype of their child’s ADHD diagnosis, so we do not have accurate information regarding this aspect; (3) parents reported whether the child was diagnosed at some point during development (at an average age of 8 years) but we do not know the updated diagnostic status at adolescence. Given these problems, and since one of our aims was to examine specific relations with the different symptom domains, we did not analyze the results with the parent-report ADHD diagnosis variable. We do mention the proportion of adolescents who were diagnosed with ADHD in the Participants section (line 210).
  • We agree that the term “nonclinical sample” might indeed be misleading, and it has been changed to “a sample of adolescents at varying levels of familial risk for ADHD” (line 517).
  • It is also important to mention that in the initial phases of the study when the children were very young, the analyses were based on a crude dichotomic split of risk and comparison groups. These analyses formed the bases of some of our early publications. However, in all our longitudinal analyses in which we tested the prediction of continuous outcomes, we used the continuous parental symptoms scores as a measure of risk (e.g., Berger et al.,  2013; Auerbach et al., 2017). We have explained this more clearly in lines 198-208.
  • The sentence regarding the clinical implications was toned-down; see line 24.
  1. Stop task ERPs

Focusing hypotheses on the sucessful stop N2 is fine, but effects on other components, particularly the STOP-P3 but also earlier activity preceding the stop signal, which are also implicated in ADHD in most papers cited (including their own) are needed to support specific interpretions.

  • Thank you for the comment. We have more clearly justified the focus on the N2; see lines 83-94.

The ERP illustrations are also not sufficient to allow judging basic ERP data quality and the presence of typical ERP tpographies and effects.

  • Thank you for the important feedback, we have improved the visualization of the results and include topographic maps; see lines 419 and 473.

Details

  • As child ADHD Symptoms at age 54 months, 7 years, and 17 years (i.e current ADHD symptoms) were averaged they contain predictors (implying temporal precedence) but also imple correlates of the N2 at age 17.
    • Following this suggestion, we separated concurrent symptoms from childhood symptoms and updated the Results section, accordingly, see section 3.4 in line 403.
  • Table 1 -check font size
    • The font size was corrected.
  • Please spell out inhibitory control (IC is not a common abbreviation in this field)
    • We have now spelled out inhibitory control throughout the MS.

Reviewer 2 Report

According to section 2.3.2: The section describes the process of EEG acquisition and preprocessing well. All the necessary details are given, and the procedures follow the standards. Only the reasoning behind selection of 40 Hz low-pass filter is missing (was it needed?). Minor in line 288: „k” instead of „K According to section 3: Table 3. would be easier to read if transposed or if the first row with numbers is moved to the last row. Also, if the numbers in the first column are all aligned to the left. Additionally, the explanation of the + sign is missing. Minor in line 408: missing dot for the beta value

Author Response

Response to Reviewers

We are very grateful to be given the chance to revise and resubmit this manuscript. We thank the editor and the reviewers for their thoughtful and helpful comments. We have revised the manuscript based on their suggestions.

Reviewer 2:

  • According to section 2.3.2: The section describes the process of EEG acquisition and preprocessing well. All the necessary details are given, and the procedures follow the standards. Only the reasoning behind selection of 40 Hz low-pass filter is missing (was it needed?). 
    • A low pass filter is recommended to attenuate line noise and muscle activity. This is a standard filter and it was used in many previous studies, for example:
      • Luo, X., Guo, J., Liu, L., Zhao, X., Li, D., Li, H., ... & Song, Y. (2019). The neural correlations of spatial attention and working memory deficits in adults with ADHD. NeuroImage: Clinical, 22, 101728.‏
      • Berger, A., Alyagon, U., Hadaya, H., Atzaba‐Poria, N., & Auerbach, J. G. (2013). Response Inhibition in Preschoolers at Familial Risk for Attention Deficit Hyperactivity Disorder: A Behavioral and Electrophysiological Stop‐Signal Study. Child Development84(5), 1616-1632.
      • Friedl, W. M., & Keil, A. (2020). Effects of Experience on Spatial Frequency Tuning in the Visual System: Behavioral, Visuocortical, and Alpha-band Responses. Journal of Cognitive Neuroscience32(6), 1153-1169.
      • Liao, X., Yao, D., Wu, D., & Li, C. (2007). Combining spatial filters for the classification of single-trial EEG in a finger movement task. IEEE transactions on biomedical engineering54(5), 821-831.‏

  • Minor in line 288: „kΩ” instead of „KΩ” 
    • Thank you, this was corrected
  • According to section 3: Table 3. would be easier to read if transposed or if the first row with numbers is moved to the last row. Also, if the numbers in the first column are all aligned to the left. Additionally, the explanation of the + sign is missing.
  • Minor in line 408: missing dot for the beta value
    • Thank you. We corrected format issues.

Reviewer 3 Report

In this prospective longitudinal study, precursors/early predictors of the N2 during response inhibition and response inhibition itself (at age 17) were measured.  What the authors found was that children’s effortful control at age 3 and parental inattention predicted N2 at age 17, but not SSRT (behavioral measure of response inhibition). While inattention throughout development predicted N2, this association did not survive after adding the early precursors to the model.

I think the manuscript is generally well written and the rationale is clear. The finding that parental inattention and effortful control at age 3 predict a neural correlate of inhibitory control at age 17 are interesting. I have the following comments:

Introduction

While the rationale is quite clear, I was initially confused because the longitudinal study in which these data were collected had as an aim to identify early predictors of ADHD symptoms. In the current study, however, ADHD symptoms are not the outcome measure but rather the predictor of the later measurement of SSRT and N2. This can be clarified better in the introduction, I believe. Specifically, can the authors more explicitly state what the relevance is of knowing precursors of reduced N2 amplitude in 17 year olds?

The authors state “Therefore, it is unclear whether these early precursors could uniquely contribute to the prediction of IC, above and beyond the development of ADHD symptoms”. What exactly is the relevance of this very questions, and how does it relate to theoretical models? For example, it may seem more logical (to me it does) to assume that early precursors may lead to weak IC which in turn may lead to symptoms of ADHD. The notion that early predictors lead to ADHD symptoms which in turn lead to weak IC, seems less logically related to explanatory models of ADHD. Can the authors clarify this more?

Methods-Results

One of the strengths of this study design lies in the longitudinal, repeated measures, design. I think it is very rich that ADHD symptoms were measured at 54 months, 7 years, and 17 years. Therefore, I thought it was a pity that the ADHD symptom scores were lumped together over these timepoints. Also, it makes it impossible to know whether really inattention symptoms OVER DEVELOPMENT predict IC at age 17, or whether the correlation between inattention symptoms and IC is driven by the concurrent correlation between inattention at age 17 and IC at age 17 (after all, the correlations between inattention at the various timepoints were small).

When inspecting the mean RT on go trials (772 ms), it occurred to me that this is a very slow reaction time, and it suggests that participants engaged in strategic slowing. This makes me wonder whether the subtraction that was used for the N2 (successful stop – slow go) may reflect strategic slowing rather than inhibition alone? It also makes me wonder how reliable SSRT is, given the strategic slowing that was likely going o

Discussion

My main questions relate to the interpretation and implications of the findings. Generally, I find that the discussion contains quite a lot of restating the findings, and less room for interpretations, meaning, and implications. Specifically:

What is the meaning of the predictive effects of early factors on N2, in the absence of such an effect on the behavioral measure of IC, SSRT?

“Our study also provides evidence that inattention symptoms in childhood, and its consistency throughout development, can longitudinally predict this electrophysiological response associated with IC in adolescence” – can this really be concluded given that inattention was lumped together across timepoints and the correlation with IC could have been driven by inattention at age 17 only.

“In fact, the contribution of child symptoms throughout development to the prediction of the N2 amplitude was no longer significant; inattention symptoms did not explain a unique additional part of the variance, above and beyond the part that was already explained by the early precursors” – what does this MEAN? What are the implications of this for theory, for clinical practice? For how we think about the development of psychopathology over time? Etc.

“The direct effects suggest that the genetic and environmental risk, inherent in paternal symptomatology and child temperament, can longitudinally predict a dysfunction in the neural mechanism of inhibition, regardless of whether the child exhibited behavioral symptoms of ADHD throughout development.” – this is more of a restatement of the findings than a discussion. Also here, I wonder whether the authors can discuss what this means for theory, our thinking of how psychopathology develops over time, about the causal relations between certain neural substrates and symptoms, etc.

“Our findings have plausible clinical implications because they may contribute to early identification of children at familial and temperamental risk of a later IC dysfunction and ADHD.” What exactly are these clinical implications?

Author Response

Response to Reviewers

We are very grateful to be given the chance to revise and resubmit this manuscript. We thank the editor and the reviewers for their thoughtful and helpful comments. We have revised the manuscript based on their suggestions.

Reviewer 3:

In this prospective longitudinal study, precursors/early predictors of the N2 during response inhibition and response inhibition itself (at age 17) were measured.  What the authors found was that children’s effortful control at age 3 and parental inattention predicted N2 at age 17, but not SSRT (behavioral measure of response inhibition). While inattention throughout development predicted N2, this association did not survive after adding the early precursors to the model.

I think the manuscript is generally well written and the rationale is clear. The finding that parental inattention and effortful control at age 3 predict a neural correlate of inhibitory control at age 17 are interesting. I have the following comments:

Introduction

While the rationale is quite clear, I was initially confused because the longitudinal study in which these data were collected had as an aim to identify early predictors of ADHD symptoms. In the current study, however, ADHD symptoms are not the outcome measure but rather the predictor of the later measurement of SSRT and N2. This can be clarified better in the introduction, I believe. Specifically, can the authors more explicitly state what the relevance is of knowing precursors of reduced N2 amplitude in 17 year olds?

  • Thank you for your comment. We have now improved the rationale of testing the inhibitory control measures as the outcome, see lines 99-117 and clarified it in The Present Study section, lines 165-166.

The authors state “Therefore, it is unclear whether these early precursors could uniquely contribute to the prediction of IC, above and beyond the development of ADHD symptoms”. What exactly is the relevance of this very questions, and how does it relate to theoretical models? For example, it may seem more logical (to me it does) to assume that early precursors may lead to weak IC which in turn may lead to symptoms of ADHD. The notion that early predictors lead to ADHD symptoms which in turn lead to weak IC, seems less logically related to explanatory models of ADHD. Can the authors clarify this more?

  • Thank you for pointing out that the study aims were not described in a clear enough way. We aimed to identify early precursors of an inhibitory control deficit in adolescence. To examine whether these precursors were indeed related to inhibitory control and that their relation with inhibitory control was not derived from the overlap of inhibitory control with ADHD symptomatology, we needed to control for ADHD symptoms. We did not examine how this deficit develops nor aimed to determine the causal direction with ADHD. We have now better explained this in lines 156-163 of the Introduction. We also clarified this in the Discussion section, lines 612-625.

Methods-Results

One of the strengths of this study design lies in the longitudinal, repeated measures, design. I think it is very rich that ADHD symptoms were measured at 54 months, 7 years, and 17 years. Therefore, I thought it was a pity that the ADHD symptom scores were lumped together over these timepoints. Also, it makes it impossible to know whether really inattention symptoms OVER DEVELOPMENT predict IC at age 17, or whether the correlation between inattention symptoms and IC is driven by the concurrent correlation between inattention at age 17 and IC at age 17 (after all, the correlations between inattention at the various timepoints were small).

  • Thank you for this important comment. Following this and also the similar comment of Reviewer 1, we have separated symptoms into concurrent (17 years) and childhood symptoms (54 months and 7 years), see lines 240-257. In section 3.4 we tested the contribution of ADHD symptoms to the SSRT and N2 separately for concurrent symptoms (lines 405-417) and childhood symptoms (lines 425-441). The path analysis was also constructed with concurrent and childhood symptoms separately (lines 447-466).

When inspecting the mean RT on go trials (772 ms), it occurred to me that this is a very slow reaction time, and it suggests that participants engaged in strategic slowing. This makes me wonder whether the subtraction that was used for the N2 (successful stop – slow go) may reflect strategic slowing rather than inhibition alone? It also makes me wonder how reliable SSRT is, given the strategic slowing that was likely going on

  • Thank you very much for noticing this. We checked the descriptive statistic and there was a typo in the mean RT of go trials, which is now corrected. The mean RT is 678.5 (see line 291). Comparing this RT with mean RTs found in previous SST adolescent studies revealed that the RT in our sample was very reasonable (the range for adolescents in control groups was approximately 670-950 and for ADHD group 625-870; Johnstone et al., 2007; Liotti et al., 2010; Liotti et al., 2007; Liotti et al., 2005; Tillman et al., 2007, see the table below).
  • Regarding the slowing strategy, we agree that when the participants can predict the occurrence of the stop signal, they might also wait for it. We took the following measures to minimize slowing, as also recommended by Verbruggen et al. (2019): first and most importantly, we used a staircase dynamic-tracking procedure which adjusted task difficulty by changing the stop-signal delay based on adolescents’ performance: SSD increases after each successful stop trial and decreases after each unsuccessful stop. By using this tracking procedure, we obtained a broad range of stop-signal delays, which helped to prevent, to some extent, the ability to strategically slow the reaction, as it is difficult to predict the occurrence of the stop signal, and therefore to wait for it. Also, as recommended by Verbruggen et al. (2019), participants were instructed to respond as quickly as possible to the go signal and not to wait for the stop signal to occur (this was added to line 264); we also explained that they will be able to stop their responses in approximately 50% of the stop trials. Moreover, during practice, the experimenter (a PhD student) was in the room to ensure that participants understood the instructions. Participants were also reminded about the instructions between blocks. Based on these measurements taken to minimize slowing as well as the corrected mean RT, we do not believe that there is a problem of slowing strategy (specific participants whose performance did not meet the model assumptions, in part because of such a strategic slowing, were removed from further analyses).
  • We have now more clearly explained the rationale for using difference wave (lines 327-340). Regarding the reliability of the SSRT, as mentioned, we analyzed the task according to Verbruggen et al. (2019) guidelines. First, we check the assumptions of the horse race model and did not estimate the SSRT when they were violated. Second, based on their stimulation results, we chose to estimate the SSRT using the integration method (with the replacement of go omissions), which is a more reliable estimation of the SSRT. We have added the following sentence to line 276 “The SSRT was calculated with the integration method with the replacement of go omissions; this method for the estimation of the SSRT was found to be more reliable and less biased, as compared to the more traditional ‘mean method’, especially when combined with a tracking procedure”.

Mean RT of go trials in previous studies in the field of ADHD:

ADHD

control

Liotti et al., 2010

children and adolescents

839.6 (175.9)

944.9 (157.2)

Liotti et al., 2007

adolescents

865.5 (164.1)

966.3 (147.4)

Liotti et al., 2005

adolescents

625 ms (140)

679 ms (114)

Johnstone et al., 2007

adolescents

742.6 (116.5) for the combined subtype and 689.7 (143.2) fot the inattentive subtype and

666.8 (162.2)

Tillman et al., 2007

adolescents

700.58 (138.44)

Discussion

My main questions relate to the interpretation and implications of the findings. Generally, I find that the discussion contains quite a lot of restating the findings, and less room for interpretations, meaning, and implications. Specifically:

What is the meaning of the predictive effects of early factors on N2, in the absence of such an effect on the behavioral measure of IC, SSRT?

  • First, it should be noted that after separating concurrent and childhood symptoms, we did find that childhood symptoms were predictive of the SSRT (see lines 435-439). It should be considered that this was a small effect and that we could not estimate the SSRT for 14 participates, which led to lower statistical power for the analyses that included the SSRT. Regarding why the early risk factors were predictive of the N2 and not of the SSRT, we believe that the SSRT has less sensitivity (compared to electrophysiological measures) for detecting subtle individual differences in a sample of adolescents at varying levels of risk for ADHD (as compared to the more prominent differences which are usually found when examining diagnosed individuals versus control participants). We discussed this in lines 626-636.

“Our study also provides evidence that inattention symptoms in childhood, and its consistency throughout development, can longitudinally predict this electrophysiological response associated with IC in adolescence” – can this really be concluded given that inattention was lumped together across timepoints and the correlation with IC could have been driven by inattention at age 17 only.

  • As concurrent symptoms were now tested separately from childhood symptoms, this sentence also was changed accordingly, see line 515.

“In fact, the contribution of child symptoms throughout development to the prediction of the N2 amplitude was no longer significant; inattention symptoms did not explain a unique additional part of the variance, above and beyond the part that was already explained by the early precursors” – what does this MEAN? What are the implications of this for theory, for clinical practice? For how we think about the development of psychopathology over time? Etc.

  • Thank you very much for this important comment, we improved this section and discussed this in lines 579-596.

“The direct effects suggest that the genetic and environmental risk, inherent in paternal symptomatology and child temperament, can longitudinally predict a dysfunction in the neural mechanism of inhibition, regardless of whether the child exhibited behavioral symptoms of ADHD throughout development.” – this is more of a restatement of the findings than a discussion. Also here, I wonder whether the authors can discuss what this means for theory, our thinking of how psychopathology develops over time, about the causal relations between certain neural substrates and symptoms, etc.

  • We have changed the structure of the discussion and removed this sentence to avoid The meaning of the direct effects is now explained in lines 579-596. We also improved the discussion about our interpretation of the relation between temperamental effortful control and the N2 and have explained that this reflects the consistency of deficits in self-regulatory skills, see lines 559-578.

“Our findings have plausible clinical implications because they may contribute to early identification of children at familial and temperamental risk of a later IC dysfunction and ADHD.” What exactly are these clinical implications?

  • We toned-down this sentence, see line 664.

Reviewer 4 Report

Sixty-three boys aged 17 who were followed since birth as part of a longitudinal study of ADHD developmental pathways completed a stop-signal task while EEG was recorded.  Reaction times and the amplitude of an N2 difference wave (averaged from 6 electrodes in the right inferior frontal cortex region) during successful trials were analyzed. Analyses indicated that higher inattention symptoms predicted reduced N2 amplitude. Familial risk for ADHD (father’s inattention symptoms), and early childhood temperament (at 36 mo of age) predicted N2 amplitude after controlling for consistency of inattention symptoms throughout development. The results might have implications for identifying children at risk of ADHD and IC dysfunction in adolescence.

Overall, this appears to be a competent study. The following items should be addressed:

  1. L202 assess (not assessed)
  2. L236 Inattention symptoms and hyperactivity-impulsivity symptoms from mothers and adolescents were averaged –but correlations were only .35 and .23. Please clarify why averaged scores were used instead of using the individual predictors (symptoms rated by mothers; symptoms rated by adolescents). Wouldn’t using one or the other provide more utility as a predictor in practice?
  3. Please explain why such a high number of STOP trials were rejected and comment on how this affects reliability of results.
  4. Related to #3: The mean number of STOP trials was 35.16. What was the lowest number for any participant?
  5. Please provide more rationale for using the difference wave (Successful STOP ERP – Successful GO), and clarify in results and discussion that the N2 was measured from the difference wave.
  6. L317 says Grand average ERP is shown in Fig.1. Please clarify that this is the difference wave. Also, show grand average raw waveforms for STOP and GO trials in addition to the difference wave in Fig. 1. 
  7. What filtering was used for the waveform in Fig. 1?
  8. In Fig. 1, the circled “region of interest” contains 13 electrode sites but only 6 sites were averaged. Please show the sites used for the average.
  9. L328 states control variables were entered in Step 1—but if it was only Mother’s education (as listed in Table 4), just indicate this.
  10. Table 3. I’m not seeing what the + sign after Mother’s inattention symptoms denotes in the Note
  11. L383, change wording here and elsewhere: analyses are conducted; models are constructed, not conducted.

Author Response

Response to Reviewers

We are very grateful to be given the chance to revise and resubmit this manuscript. We thank the editor and the reviewers for their thoughtful and helpful comments. We have revised the manuscript based on their suggestions.

Reviewer 4:

Sixty-three boys aged 17 who were followed since birth as part of a longitudinal study of ADHD developmental pathways completed a stop-signal task while EEG was recorded.  Reaction times and the amplitude of an N2 difference wave (averaged from 6 electrodes in the right inferior frontal cortex region) during successful trials were analyzed. Analyses indicated that higher inattention symptoms predicted reduced N2 amplitude. Familial risk for ADHD (father’s inattention symptoms), and early childhood temperament (at 36 mo of age) predicted N2 amplitude after controlling for consistency of inattention symptoms throughout development. The results might have implications for identifying children at risk of ADHD and IC dysfunction in adolescence.

Overall, this appears to be a competent study. The following items should be addressed:

     1. L202 assess (not assessed)

  • Thank you, this was corrected

    2. L236 Inattention symptoms and hyperactivity-impulsivity symptoms from mothers and adolescents were averaged –but correlations were only .35 and .23. Please clarify why averaged scores were used instead of using the individual predictors (symptoms rated by mothers; symptoms rated by adolescents). Wouldn’t using one or the other provide more utility as a predictor in practice?

  • Based on your comment we conducted the analyses with self and mother reports separately; indeed, the variable of mother reports provided more utility as a predictor, so we used it instead of using the combined report. It is also important to mention that, as mentioned above, following the reviewers' suggestions we separated concurrent and childhood symptoms and conducted all analyses with these separated scores. For simplicity, we removed the concurrent self-reports from the manuscript (otherwise, we need to report all results with childhood symptoms, concurrent mother reports, and concurrent self-report, separately for inattention and hyperactivity, and this may be confusing for readers). However, we are willing to include it upon request.

     3. Please explain why such a high number of STOP trials were rejected and comment on how this affects reliability of results.

  • See the answer to question 4

     4. Related to #3: The mean number of STOP trials was 35.16. What was the lowest number for any participant?

  • The number of successful stop trials was not the result of a major rejection of data; it was based on the design of the task. We used a staircase dynamic-tracking procedure (Logan et al., 1994), which adjusted task difficulty by changing the stop-signal delay (SSD) based on adolescents’ performance. The algorithm was programmed to lock on the stop-signal delay (SSD), which produced approximately 50% successful inhibition trials. Therefore, from a total number of 72 stop trials, there was a mean of 35 successful trials. The number of correct stop trials ranged from 23-49 and this information was added to line 325 (the quartiles for the correct number of trials were as follow, Q1=23-33, Q2=33-34, Q3=34-36.5, and Q4=36.5-49). It should be noted that a similar number of successful stop trials were used in previous studies for the calculation of the N2 amplitude; see for example a mean of 26.3 successful stop trials in Johnstone and colleagues (2007) and a minimum of 20 trials in Senderecka and colleagues’ (2012) study.
    • Johnstone, S. J., Dimoska, A., Smith, J. L., Barry, R. J., Pleffer, C. B., Chiswick, D., & Clarke, A. R. (2007). The development of stop-signal and Go/Nogo response inhibition in children aged 7–12 years: performance and event-related potential indices. International Journal of Psychophysiology, 63(1), 25-38.
    • Senderecka, M., Grabowska, A., Szewczyk, J., Gerc, K., & Chmylak, R. (2012). Response inhibition of children with ADHD in the stop-signal task: An event-related potential study. International Journal of Psychophysiology, 85(1), 93-105.
  • We have now calculated the split-half reliability for the N2 amplitude which was 0.77 (this information was added to line 340)

     5. Please provide more rationale for using the difference wave (Successful STOP ERP – Successful GO), and clarify in results and discussion that the N2 was measured from the difference wave.

  • We have elaborated the rationale for using the difference wave in lines 327-340.
  • We clarified that the N2 was measured from the difference wave in all figures (see lines 342,419, 473, and in the discussion, line 495).

     6. L317 says Grand average ERP is shown in Fig.1. Please clarify that this is the difference wave. Also, show grand average raw waveforms for STOP and GO trials in addition to the difference wave in Fig. 1. 

  • We have added 2 figures and explain that the plot represents the difference wave (line 420 and 473). As requested, we added the raw waveforms for stop and go trials to Figure 1 (line 341).

    7. What filtering was used for the waveform in Fig. 1?

  • We did not use additional filtering besides what was described in the pre-processing of EEG data (line 310).
  •  
  • 8. In Fig. 1, the circled “region of interest” contains 13 electrode sites but only 6 sites were averaged. Please show the sites used for the average.
  • Thank you for noticing; we corrected the circle and it now includes the relevant 6 electrodes; see Figure 1, line 341.
  •  
  • 9. L328 states control variables were entered in Step 1—but if it was only Mother’s education (as listed in Table 4), just indicate this.

  • The sentence now states: “The control variable of mother’s education was entered in step 1” (line 352).

     10. Table 3. I’m not seeing what the + sign after Mother’s inattention symptoms denotes in the Note

  • We have added +p < .10 to the notes of table 3, line 402.

     11. L383, change wording here and elsewhere: analyses are conducted; models are constructed, not conducted.

  • Thank you; this was changed.

Round 2

Reviewer 1 Report

The authors have done a good job clarifying the open issues regarding ADHD status and resolving many of my concerns.

Still the low rate of T-scores that indicating ADHD symptom in the clinical or subclinical range (10% and 5% as reported now) should also explicitly be added as a limitation for conclusions regarding the risk for ADHD as a disorder.

I still find it a pity to focus only on the Stop-N2, since Fig. 1 suggests a clear effect of inattention on the subsequent positivity,  but I leave this to the  authors (they may possibly want to extend this focus in a  subsequent paper). I understand that the N2-focus can be justified by a previous lack of correlation between ADHD-scores and the subsequent positivity at age 5y,  but this was for a small subsample at a different age.

Author Response

  • Thank you for the positive feedback.
  • We added the following sentence to the limitation section (line 665-668): It should also be noticed that according to mother’s reports of adolescents’ ADHD symptoms, there was a low rate of T-scores in the clinical or subclinical range. This limits to some degree our ability to generalize our conclusions regarding the risk for ADHD as a disorder”.
  • To comply with the time limitations we could not add the P3 to the current paper. However, this is an ongoing project, and we continue to explore other ERP components and additional electrophysiological signatures, such as the power of theta frequency, intrasubject variability, and inter-trial coherence, etc.